# The Streptococcus agalactiae R3 surface protein is encoded by *sar5*

**Adelle Basson**[1], **Camilla Olaisen**[2], **Linn-Karina Selvik**[1,3], **Randi Valsø Lyng**[2], **Hilde Lysvand**[1], **Alexandre Gidon**[1,3], **Christina Gabrielsen Aas**[1,2], **Jan Egil Afset**[1,2☯], **Marte Singsås Dragset**[1,3☯] *

**1** Department of Clinical and Molecular Medicine, Norwegian University of Science and Technology (NTNU), Trondheim, Norway, **2** Department of Medical Microbiology, St. Olavs University Hospital, Trondheim, Norway, **3** Centre for Molecular Inflammation Research (CEMIR), Norwegian University of Science and Technology (NTNU), Trondheim, Norway

☯ These authors contributed equally to this work.
* marte.dragset@ntnu.no

**Data Availability Statement:** All relevant data are within the paper and its Supporting Information files.

**Funding:** This work was supported by the Research Council of Norway through its Centres of

## Abstract

Streptococcus agalactiae (group B streptococcus; GBS) is an important human pathogen causing pneumonia, sepsis and meningitis in neonates, as well as infections in pregnant women, immunocompromised individuals, and the elderly. For the future control of GBS-inflicted disease, GBS surface exposed proteins are particularly relevant as they may act as antigens for vaccine development and/or as serosubtype markers in epidemiological settings. Even so, the genes encoding some of the surface proteins established as serosubtype markers by antibody-based methods, like the R3 surface protein, are still unknown. Here, by examining a Norwegian GBS collection consisting of 140 strains, we find that R3 protein expression correlates with the presence of the gene *sar5*. By inducible expression of *sar5* in an R3-negative bacterial strain we show that the *sar5* gene product is specifically recognized by an R3 monoclonal antibody. With this we identify *sar5* as the gene encoding the R3 surface protein, a serosubtype marker of hitherto unknown genetic origin.

## Introduction

*Streptococcus agalactiae* (group B streptococcus; GBS) is an important human pathogen, most notably in neonates, but also in pregnant women as well as immunocompromised and elderly individuals. Globally, an estimated prevalence of maternal rectovaginal GBS colonization is 17.9%, with the highest and lowest mean prevalence found in Africa (22.4%) and Southeast Asia (11.1%), respectively [1]. Colonization of GBS during pregnancy is a risk factor for preterm birth, stillbirth, and neonatal infection [2]. To reduce the risk of vertical transmission of GBS to the neonate during birth, routine screening for GBS colonization followed by intrapartum antibiotic prophylaxis (IAP) to pregnant women with GBS is recommended [3]. However, administration of IAP poses a risk of allergic and anaphylactic reactions [4, 5], and the widespread use of antibiotics may result in the emergence of antibiotic resistance. Another option to prevent GBS infection is vaccine development. Currently, conserved GBS surface proteins are considered as promising targets for vaccine development [6], as they may elicit a strong immune response against the majority of GBS strains [7].

Excellence scheme, project number 223255/F50, as well as internal funds from Department of Clinical and Molecular Medicine, Norwegian University of Science and Technology (NTNU), Trondheim, Norway, and Department of Medical Microbiology, St. Olavs University Hospital, Trondheim, Norway. There was no additional external funding received for this study.

**Competing interests:** The authors have declared that no competing interests exist.

GBS surface proteins also play an important role as serosubtype markers, relevant for GBS classification in epidemiological settings. While GBS strains can be distinguished into ten serotypes due to differences in their capsular polysaccharide (CPS) (Ia, Ib, and II–IX), surface-expressed protein antigens enable further division of these serotypes. Some of the surface proteins are conserved and present in nearly all GBS strains, while others are associated with specific serotypes, and thus used to define serosubtypes [8]. Historically, detection of serosubtypes by means of antibody-based methods has played a major role. In more recent years, serosubtyping of GBS has benefitted greatly from the introduction of molecular methods, such as PCR and whole genome sequencing (WGS) [9, 10].

GBS surface proteins have been classified according to two different and overlapping classification systems (Table 1). However, there is still some discrepancy and confusion surrounding the traditional nomenclature, and some surface proteins that have not yet been definitely linked to a specific gene. One classification scheme of GBS surface proteins includes Cβ and the Cα-like proteins (Alps) Cα, Alp1-4 and Rib. Nearly all GBS strains carry one of the six alp genes (Alp GBS) although, occasionally, an Alp-encoding gene may be absent (non-Alp GBS) [11]. Another, and overlapping, classification system of GBS surface proteins is the Streptococcal R proteins first described in 1952 [12], which are resistant to trypsin digestion (thereby designated "R"). R proteins are categorized into five types, R1-5 [13–15]. R1 is probably non-existent as a distinct protein; the antiserum raised against R1 was later shown to recognize the identical N-termini of Alp2 and Alp3, the gene products of *alp2* and *alp3*, respectively [16]. The R2 protein is expressed by group A and C streptococci and does not seem to occur in GBS [15]. The R4 protein has been shown to be identical to Rib and is encoded by the *rib* gene [17], while R5 has been renamed group B protective surface protein (BPS) and was shown to be the gene product of *sar5* [15, 18]. The R3 protein has been characterized to some extent [14, 19–21], and has proved useful as a serosubtype GBS marker [22, 23]. However, the gene encoding the R3 protein is still unknown (Table 1). BPS was initially thought to be distinct from R3 [15], however, a later study pinpointed a correlation between the presence of the BPS-encoding *sar5* gene and R3 expression [8]. Here, we follow up on this correlation, hypothesizing that *sar5* encodes R3. Unraveling the R3-encoding gene, and the putative discrepancy in the nomenclature and nature of the *sar5* gene product, is important for the *sar5* gene product as a prospective target in vaccine development and molecular based GBS serosubtyping, as well as for functional studies on its mechanistic role in pathogenicity.

**Table 1. Surface-proteins of GBS.**

| Name | Gene | GenBank Number |
|------|------|----------------|
| **Cα** | *bca* [24] | M97256 |
| **Alp1 (epsilon)** | *alp1* [25] | AH013348.2 |
| **Alp2**/R1 | *alp2* [16, 26] | AF208158 |
| **Alp3**/R1 | *alp3* [16, 26] | AF245663 |
| **Alp4** | *alp4* [27] | AJ488912 |
| R3 | unknown[a] | - |
| **Rib**/R4 | *rib* [17] | U583333 |
| R5/BPS | *sar5* [15] | AJ133114 |

Alps (in bold) and R proteins.

[a] in this study determined to be encoded by *sar5*.

## Results

### Presence of the *sar5* gene correlates with R3 protein expression across GBS strains

In a previous study, 121 GBS strains collected from pregnant women in Zimbabwe were tested for (among other markers) the presence of the *sar5* gene and R3 protein expression [8]. The study found that 31 out of 35 (91.5%) *sar5* positive strains expressed R3. The remaining 86 strains were negative for both *sar5* and R3. Based on these findings we speculated that *sar5* could encode R3, and that, consequently, the previously reported *sar5*-encoded BPS and the R3 protein are the same protein. To further investigate this observed association between *sar5* and R3 expression, we analyzed 140 clinical GBS strains from neonatal and adult GBS infections from the Norwegian GBS reference laboratory [20]. These strains were previously characterized for R3 expression (and other serotype markers) by fluorescent antibody testing using a monoclonal R3 antibody [20]. This R3 antibody has been used and evaluated in several previous studies [8, 11, 22, 23, 28, 29]. We typed the strains for presence/absence of *sar5* using a previously established PCR approach [8], with the R3 reference strain CCUG 29784 (also known as Prague 10/84) as a positive control. Of the 140 GBS strains, the majority were *sar5* negative (131), while nine strains were *sar5* positive. Seven of these strains were R3 positive (Table 2, S1 Fig and S1 Table). For three representative strains (the R3 reference strain CCUG 29784, the R3-negative *sar5*-positive strain 93–33, and the R3-positive *sar5*-positive strain 93–50) we confirmed by Sanger sequencing that the PCR products indeed represented amplification of the *sar5* gene (S2 Fig). Hence, there was a strong, albeit not perfect, correlation between the presence of the *sar5* gene and R3 expression across the 140 investigated GBS strains.

### The *sar5* positive R3 negative GBS strains express R3 but encode a *sar5* deletion variant

Two strains, 93–33 and 94–3, contained the *sar5* gene but were negative for R3 expression. Thus, these strains were in conflict with our hypothesis that *sar5* encodes R3. The initial detection of R3 in the 140 GBS strains included in this study was performed by direct fluorescent monoclonal antibody testing (DFA) on whole bacterial cells [20, 30]. To re-ensure that these two strains are indeed not recognized by the R3 antibody in a whole cell state, we R3-stained and subjected them to confocal microscopy together with the R3/*sar5* negative (CCUG 29779) or positive (CCUG 29784) control strains (Fig 1A). By quantifying their R3 antibody signals, the 93–33 and 94–3 strains showed the same signal level as the negative control (Fig 1B).

We speculated that R3 was still expressed in the *sar5*-positive R3-negative strains, but that it was not detected by antibody staining of whole cells, for instance due to lack of surface expression. We therefore subjected the strains to western blot analysis of denatured proteins from cell lysates. Indeed, using the same R3 monoclonal antibody as previously, we could detect R3 in the 93–33 and 94–3 lysates, in addition to the CCUG 29784 control lysate (Fig 1C). For all three strains the blot displayed a ladder-like pattern characteristic for the R3 protein [20]. However, the largest ladder fragment in the 93-33/94-3 lysates was smaller in size compared to

**Table 2. Distribution of the *sar5* gene and R3 expression among 140 GBS clinical strains.**

|  | R3 positive | R3 negative | Sum |
|---|---|---|---|
| *sar5* positive | 7 | 2 | 9 |
| *sar5* negative | 0 | 131 | 131 |
| **Sum** | **7** | **133** | **140** |

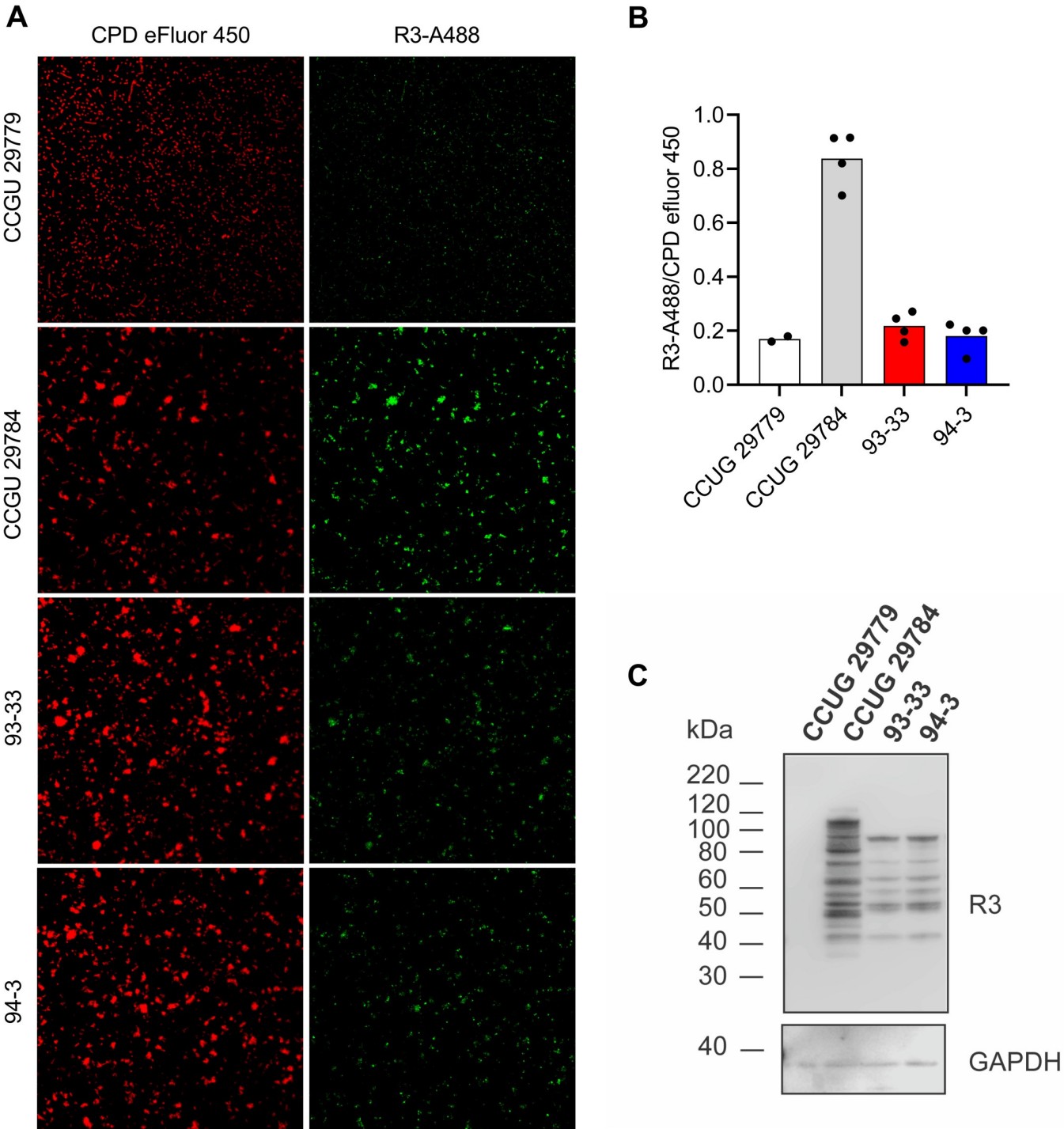

**Fig 1.** (A) Visualization of the R3 antigen at the cell surface of strains 93–33, 94–3, CCUG 29779 (negative control) and CCUG 29784 (positive control) by confocal microscopy and direct antibody staining using α-R3 as primary antibody and Alexa Fluor 488 as secondary antibody (green). Proteins in general (protein N termini and lysine residues) were labeled with CPD-eFluor 450 and used as internal normalization control (red). (B) Quantification of the R3-Alexa 488 signal relative to the CPD eFluor 450 signal. (C) Western blot analysis of GBS whole cell denatured lysates from strains 93–33, 94–3, CCUG 29779 (negative control) and CCUG 29784 (positive control). The blots were probed with α-R3 antibody (upper panel) and α-GAPDH antibody as an internal loading control (lower panel). The protein standards are shown to the left of the blots. Raw data images of western blots are found in S1 Raw images.

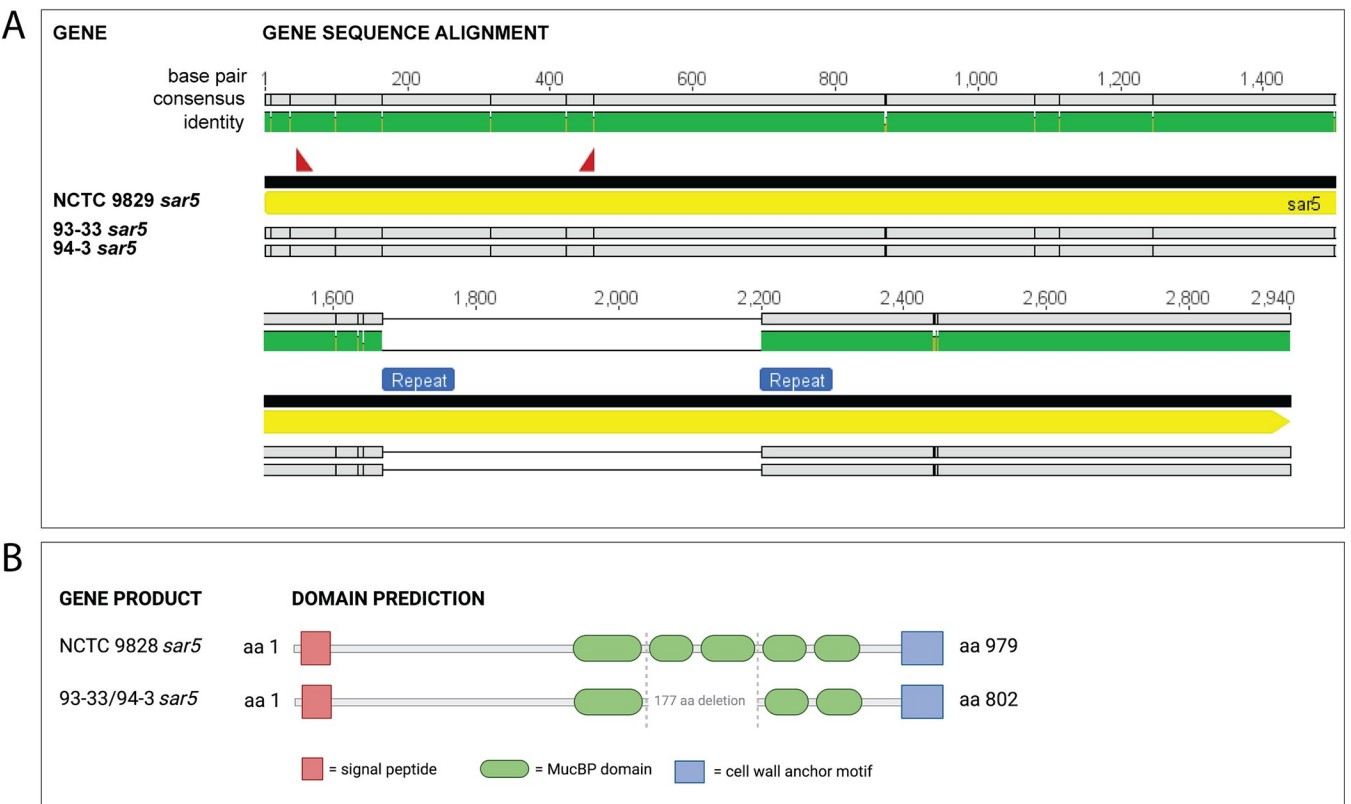

**Fig 2.** (A) Alignment of the gene sequences of strains 93–33 and 94–3 to the sar5 gene of sar5 reference strain NCTC 9828. Identity between all sequences is indicated by the top panel in green and gene annotation is shown in yellow. Black vertical lines indicate mismatches at the nucleotide level, while grey boxes indicate matching nucleotides to the reference strain. 93–33 and 94–3 have a 531 bp long deletion (marked with horizontal line) within the sar5 gene. The binding sites of the primers used to detect the sar5 gene are shown as red triangles. Repeat regions are shown in blue. (B) Pfam database prediction of protein domains and motifs of the gene products encoded by sar5 of strains NCTC 9828 and 93-33/94-3 (figure created with BioRender.com).

the control, where the largest fragment was around the expected size of 109 kDa (in accordance with the size of the *sar5* gene at 2940 bp), suggesting that the R3 expressed in 93-33/94-3 may be truncated. This prompted us to subject 93–33 and 94–3 to nanopore WGS. Indeed, both GBS strains possessed a *sar5* gene with an identical 531 bp in-frame deletion towards the 3' end of the gene, when compared to the *sar5* gene of NCTC 9828 (the *sar5* reference strain [15]) (Fig 2A). The deletion occurred between two 102 bp long direct repeat regions, making it feasible that the 531 bp (corresponding to 177 amino acids) region has been deleted by homologous recombination (Fig 2A). The 531 bp deletion corresponds perfectly to the ~20 kDa difference in size between the R3 control strain and the 93–33 and 94–3 strains observed by western blot analysis (Fig 1C).

To investigate if the deletion was likely to interfere with surface expression, we subjected the *sar5*-encoded protein sequences of strains NCTC 9828 and 93-33/94-3 to the Pfam database (https://pfam.xfam.org/) for domains/motifs prediction. We found that the deleted region corresponds to two of five mucin-binding protein domains (MucBP; PF06458) found in the full-length protein (Fig 2B). Moreover, a signal peptide (YSIRK_signal; PF04650) and a cell wall anchor motif (LPXTG gram_pos_anchor; PF00746) were predicted at the N- and C-terminus of the *sar5* gene products, respectively, for both NCTC 9828 and 93-33/94-3 (Fig 2B). These findings suggest that the deletion does not directly interfere with surface expression.

Taken together, our results show that the two *sar5* positive but initially R3 negative strains indeed express R3, although in a truncated form compared to the control strain, and that they both possess a deletion variant of the *sar5* gene.

## The *sar5*-encoded protein is recognized by the R3 antibody

Based on the above results, we had strong indications that the *sar5* gene encodes the R3 protein. We aimed to prove this experimentally by inducing *sar5* protein expression in a *sar5* negative bacterial species, followed by R3 protein detection. First, we constructed a *sar5* inducible expression vector by replacing the luciferase reporter gene of pKT1 [31] with *sar5*, creating pKT1-*sar5*-F. In pKT1, luciferase expression is controlled by the XylS/*Pm* regulator/promoter system, which is induced by the benozoic acid *m*-toluate. We added a FLAG tag to the C-terminal end of the *sar5*-encoded protein, to allow for successful detection of the *sar5*-encoded protein also if the protein was not recognized by the R3 specific antibody (Fig 3A). Even so, upon *m*-toluate induction of pKT1-*sar5*-F in *Escherichia coli* BL21 (DE3), we could clearly detect the *sar5*-encoded 109 kDa protein with the monoclonal R3 antibody (Fig 4). We could furthermore detect the FLAG-tag expressed from pKT1-*sar5*-F around the same expected size of 109 kDa, confirming that it is indeed the *sar5* gene product that is detected. When we induced pKT1 (expressing luciferase as opposed to *sar5*), we did not detect expression of any protein around 109 kDa, neither with the R3 nor the FLAG-specific antibodies.

Since the two *sar5* positive but initially R3 negative strains 93–33 and 94–3 actually expressed R3, and were shown to possess a deletion variant of *sar5* (*sar5D*), we wanted to investigate whether this deletion variant also encoded a protein which is recognized by the R3 antibody. Hence, we constructed an inducible vector expressing FLAG-tagged *sar5D*

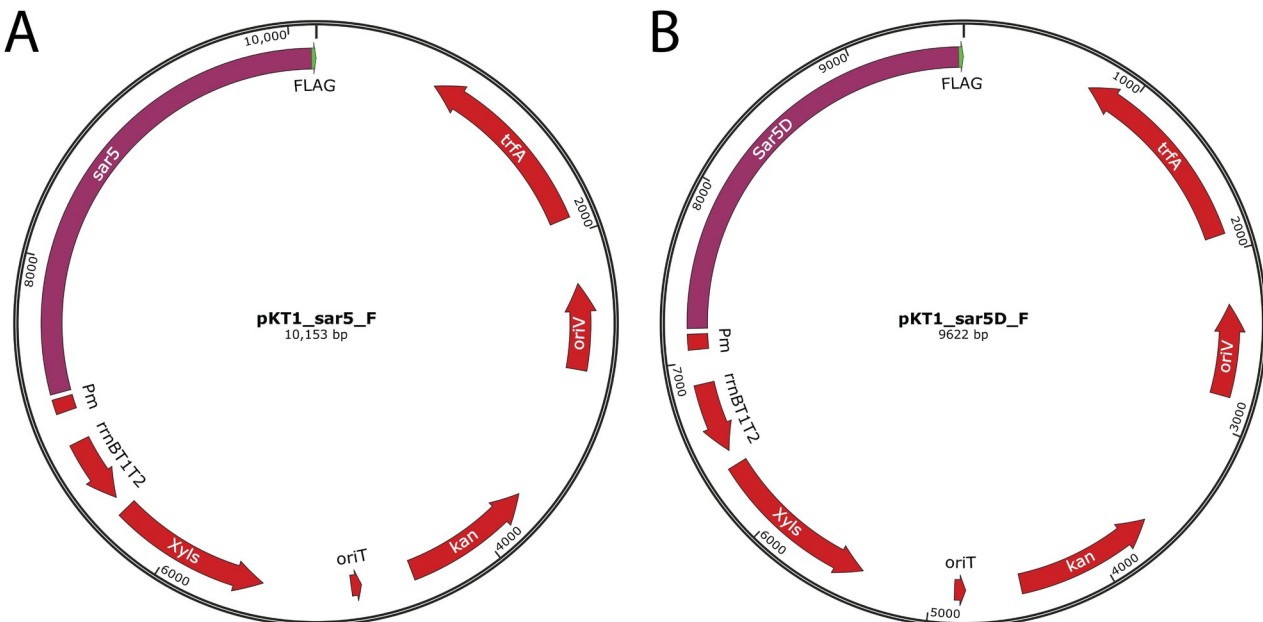

**Fig 3.** To induce expression of sar5, the luciferase reporter gene of pKT1 [31] was replaced by FLAG-tagged sar5 and FLAG-tagged sar5D, creating (A) pKT1-sar5-F and (B) pKT1-sar5D-F, respectively. xylS, gene encoding the transcription activator XylS. Pm, promoter at which XylS binds and activates transcription in response to the inducer m-toluate. kanR, gene encoding resistance to kanamycin. oriV, origin of replication for RK2-derived plasmids. trfA, gene encoding plasmid replication initiator protein TrfA, activating replication by binding to oriV. rrnBT1T2, transcriptional terminator. oriT, origin of conjugal transfer. The plasmid maps were generated using SnapGene software (from Insightful Science).

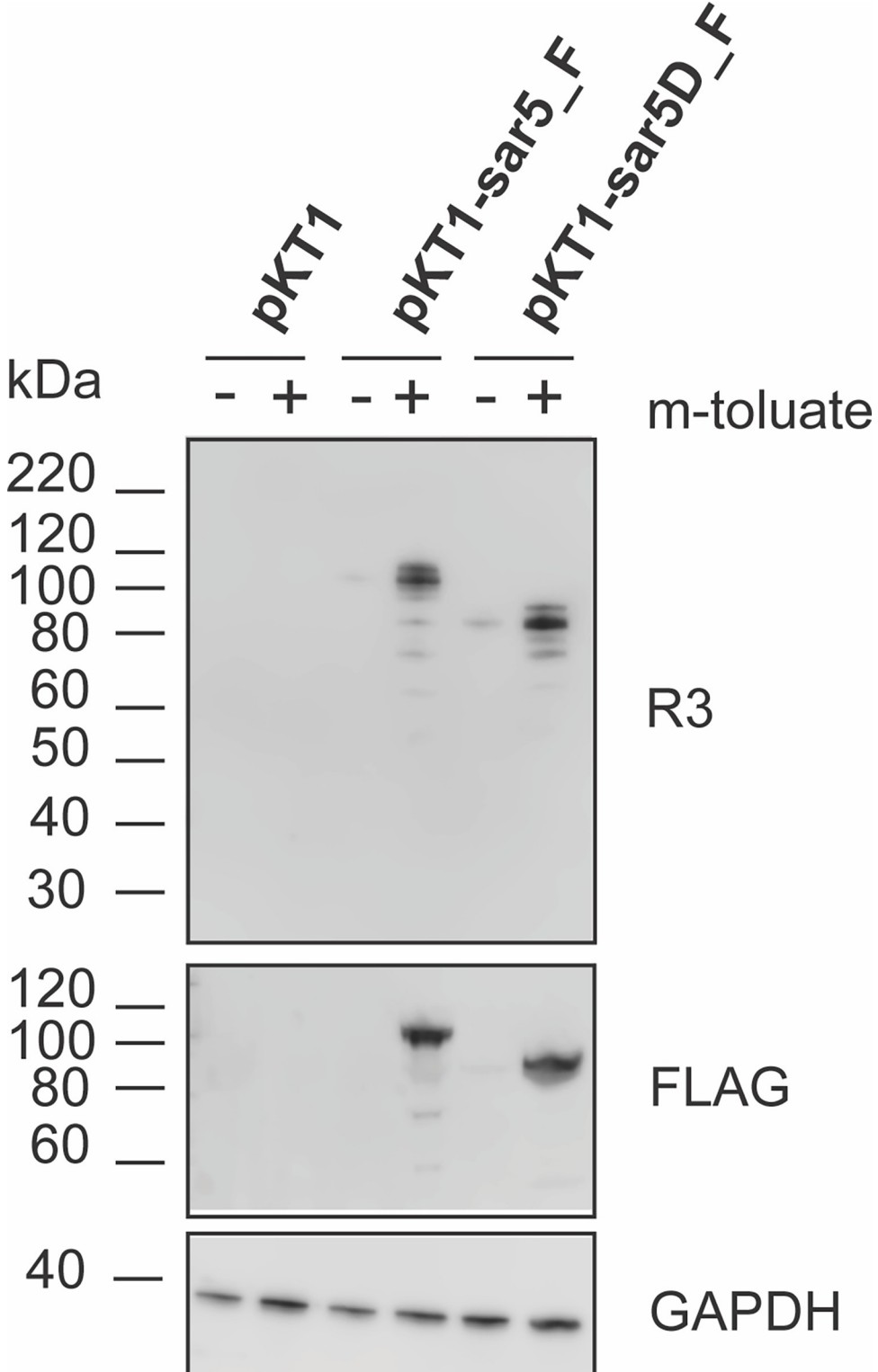

**Fig 4. Western blot analysis of whole cell lysates from E. coli BL21 (DE3) carrying pKT1, pKT1-sar5-F or pKT1-sar5D-F plasmids, induced with 2 mM m-toluate (+) or mock-induced with the equivalent amount of the solvent ethanol (-).** The blots were probed with r α-R3 antibody (upper panel), α-FLAG antibody (middle panel), or α-GAPDH antibody (lower panel). The protein standards are shown to the left of the blots. Raw data images of western blots are found in S1 Raw images.

(pKT1-sar5D-F, Fig 3B), and subjected it to induction and western blot analysis. As for the full-length *sar5*, both the R3 and the FLAG antibody bound to the induced *sar5D* gene product (Fig 4). Compared to the full-length *sar5*, the *sar5D* gene encoded a seemingly truncated R3 protein, corresponding in size to the R3 protein expressed by the 93–33 and 94–3 strains. Taken together, our results demonstrate that *sar5* encodes a protein recognized by the R3-specific antibody.

## Materials and methods

### Bacterial strains

Included in this study were 140 clinical GBS strains collected in the years between 1993 and 1995 from neonatal or adult GBS disease from the Norwegian national reference laboratory for GBS, Department of Medical Microbiology, St Olavs Hospital, Trondheim, Norway. These strains were previously characterized for R3 expression [20], as shown in S1 Table. *S. agalactiae* CCUG 29784 was used as an R3 reference strain, while *S. agalactiae* CCUG 29779 (both from Culture Collection University of Gothenburg, Sweden) was used as an R3 negative control in western blot analysis. *S. agalactiae* NCTC 9828 (also called ComptonR, [15]) was used as a reference strain for the *sar5* gene in analysis of *sar5* gene sequences. The GBS strains were cultured overnight on blood agar medium or in Todd-Hewitt broth at 35°C.

### Detection of *sar5* by PCR

Bacterial cells from a single colony were suspended in 100 μl TE-buffer and 100 μl lysis buffer (1% Triton X-100, 0.5% Tween 20, 10 mM Tris-HCl with pH 8 and 1 mM EDTA) [32]. The mixture was incubated at 95°C for 15 minutes and centrifuged at 14 500 rpm for 2 minutes before 100 μl of the supernatant was transferred to a new tube. This material was used as template in the PCR reaction, with AmpliTaq Gold DNA Polymerase with Buffer I (5U/μl; Applied Biosystems). The *sar5*-specific primers used were identical to those of Mavenyengwa et al; *sar5* forward and *sar5* reverse [8]. For three representative strains (CCUG 29784 as the positive control strain, 93–33 as an R3-negative *sar5*-positive strain, and 93–50 as an R3-positive *sar5*-positive strain) the PCR products were sequenced using the *sar5* forward primer and the Eurofins Genomics sequencing services (LightRun Tube format).

### Preparation of bacteria for confocal imaging

Overnight cultures of GBS strains CCUG29779, CCUG 29784, 93–33 and 94–3 were pelleted by centrifugation, and blocked in 500 μl PBS with 20% pooled A+ human serum provided by the blood bank at St Olav's Hospital (Trondheim, Norway) and incubated at a roller shaker for 90 minutes. The samples were washed once with PBS and resuspended in PBS with 1% A + serum and α-R3 (1:1000, mouse monoclonal from [20]) before incubation at a roller shaker for 1 hour. Samples were then washed twice with PBS with 1% A+ serum and incubated with PBS with 1% A+ serum and Alexa Fluor 488 (1:1000, donkey-anti-mouse, Invitrogen, 1:1000) at a roller shaker for 45 minutes. After washing once with PBS with 1% A+ serum and once with PBS, samples were resuspended in 1 ml PBS with 0,05% Tween and eBioscience™ Cell Proliferation Dye eFluor™ 450 (1:1000, Invitrogen) and incubated at 37°C for 15 minutes. The samples were washed once in PBS before fixed in 4% ice-cold paraformaldehyde for 10 minutes on ice. After fixations, the samples were washed three times in PBS.

## Confocal imaging

PFA-fixed bacterial culture on glass-bottomed 96 well plates were imaged with a Zeiss LSM880 confocal microscope with 40x NA = 1.4 objective (Carl Zeiss Micro-imaging Inc.). Emissions were collected using GaAsP hybride detectors. The following acquisition parameters were used: 1024*1024 pixel image size, numerical zoom set to 0.6, frame averaging 1, and 3D acquisition to collect the entire cell with a Z-stack step of 0.30 μm. eFluor450 was excited with a 458 nm Argon laser and emissions were collected through a 470–500 nm window. Alexa488 was excited with a 488nm Argon laser and emissions were collected through a 505–550 nm window. Images were analyzed and quantified with Image J (NIH) as follow: 3D stack were projected using "Sum" and converted to 8-bit. Background was estimated using the "HiLo" Lookup Table and subtracted prior to record the eFluor450 and Alexa488 raw intensity.

## DNA isolation, WGS and assembly

Bacterial cells were suspended in TE buffer and treated with proteinase K (1.5 mg/mL), lysozyme (0.5 mg/mL) and mutanolysin (250 U/mL) for 15 minutes with shaking at 37˚C, before heating at 65˚C for 15 minutes. RNAse A (2 mg/mL) was then added to the lysate. Genomic DNA was subsequently isolated using the EZ1 DNA tissue kit with an EZ1 Advanced XL instrument (Qiagen). Illumina sequencing libraries were prepared using the Nextera XT sample prep kit and sequenced on the Illumina MiSeq platform (Illumina) with 300-bp paired-end read configuration (MiSeq Reagent Kit v3). Nanopore sequencing libraries were prepared using the Rapid Sequencing Kit (SQK-RAD004) and sequenced on a minION intrument with Flongle adapter and flowcells (FLO-FLG001) (Oxford Nanopore Technologies). Raw nanopore data was basecalled using Guppy v5.0.13 and assembled using Flye v2.7. Assemblies were polished with nanopore data using Racon v1.4.20 and with Illumina.data using Pilon v1.23. Geneious vR9 was used for alignments and visualization.

## Cloning of *sar5* into an inducible expression vector

The *sar5* gene was cloned into the *m*-toluate-inducible expression vector pKT1 [31]. Briefly, the *sar5* ORF of GBS strain 13/87 (identical to the BPS-encoding gene of strain NCTC 9828 [15]) was amplified. To incorporate a C-terminal FLAG-tag, the sequence encoding the FLAG epitope (DYKDDDDK), was incorporated into the *sar5*-amplification reverse primers. In addition, the primer sets amplifying *sar5* and the pKT1 vector backbone were extended with overlaps to enable Gibson Assembly with the Gibson Assembly® Master Mix (NEB). KOD Xtreme™ Hot Start DNA Polymerase (Sigma-Aldrich) was used for PCR amplifications. Illustrations generated using SnapGene software (Insightful Science; available at snapgene.com) of the cloning strategy and the primers used are found in S1 File.

## Induced expression of *sar5*

For expression of *sar5* in *E. coli*, pKT1 (negative control), pKT1-sar5_F and pKT1-sar5D_F were transformed into *E. coli* strain BL21 (DE3) and grown in LB medium supplemented with 50 μg/ml kanamycin to stationary phase, then diluted approximately 1:500 and grown to OD600 0.05–0.1 at 37˚C. At this point, the samples were adjusted to the same OD600, induced with 2 mM of *m*-toluate (Sigma, 1 M stock solution solved in laboratory grade ethanol) and incubated for 5 hours with shaking at 30˚C. For uninduced samples the equivalent amount of ethanol was added as a mock treatment.

## Preparation of protein extracts and detection of protein expression by western blot analysis

Overnight cultures of GBS strains were pelleted by centrifugation and washed in PBS. The pellets were resuspended in 1X LDS Sample Buffer (NuPage®, Invitrogen) with 50 mM dithiothreitol (DTT) and heated for 10 minutes at 95°C. Samples were cleared for cellular debris by centrifugation. To prepare protein extracts of *E. coli*, induced (or mock induced) cultures were adjusted to $OD_{600}$ ~0.8, and pelleted by centrifugation. The pellet was resuspended in 50 μl 1x LDS Loading Buffer with 50 mM DTT, heated for 10 minutes at 70°C and sonicated 3 times for 1 minute each. Protein extracts were separated on 4–12% Bis-Tris mini protein gels (NuPage®, Invitrogen) and blotted on polyvinylidene fluoride membranes (Bio-Rad). Membranes were blocked with 1X blocking buffer (Roche) in PBS. The primary antibodies against R3 (mouse monoclonal, from [20]), FLAG (monoclonal mouse anti-FLAG M2 antibody, Sigma), and GAPDH antibody GA1R (Covalab) as well as the HRP conjugated secondary antibody goat anti-mouse (Dako) were diluted in 0.5X blocking buffer/PBS. The bound HRP-conjugated antibodies were visualized using SuperSignal™ West Femto Maximum Sensitivity Substrate (Thermo Scientific) and Odyssey Fc imaging system (Licor).

## Discussion

GBS *sar5* was previously shown to encode BPS, a protein initially described to be different from the R3 surface protein [15]. Correlation between R3 expression and the presence of the *sar5* gene was observed within a previously examined GBS strain collection, where 31 out of 35 (91.5%) *sar5* positive strains expressed R3 [8]. Similarly, frequent co-expression of the Alp protein Cα and the non-Alp Cβ protein has been observed, where 81% of the Cβ positive strains also contained Cα [33]. Even so, Cα and Cβ are encoded by two different genes; *bca* and *bac*, respectively [24, 34]. To elucidate whether this was also the case for R3 and BPS, we further investigated the correlation between *sar5* and R3 expression across 140 GBS strains from the Norwegian GBS reference laboratory. We observed a perfect correlation between the presence of the *sar5* gene and R3 expression, as well as between the absence of *sar5* and lack of R3 expression. Furthermore, when we induced *sar5* expression in a non-R3 bacterial strain, we found that the monoclonal R3 antibody [20] specifically recognized the *sar5*-encoded protein. With this, we demonstrated that the *sar5* gene encodes R3 and that, consequently, R3 and BPS must be one and the same protein.

During the initial screening of our strain collection two strains were *sar5* positive but R3 negative (93–33 and 94–3). However, while these strains were deemed negative in R3 expression by fluorescent antibody testing on whole bacterial cells (reference [20] and Fig 1B), they were positive upon western blot analysis of denatured cell lysates (Fig 1C). We also found that GBS strains 93–33 and 94–33 possessed a variant of *sar5* with a deletion (*sar5D*). We speculate that the *sar5D*-encoded protein is not recognized by the monoclonal R3 antibody due to conformational changes masking the R3 epitope of the protein in its native form, or that the protein is simply not expressed on the surface of the bacterial cells and thus only detected by immunoblotting of denatured whole cell lysates. Bioinformatic prediction of the *sar5D* gene product revealed that its signal peptide and cell wall anchoring motif were not affected by the deletion (Fig 2B), providing no direct explanation of the potential lack of surface expression. Predictions also revealed that the *sar5D* deleted region corresponds to two of five MucBP domains. MucBP domains are thought to facilitate adhesion of bacteria to the host via direct interactions with mucins within epithelial cell-secreted mucus [35, 36], shedding light on a possible function of the *sar5*-encoded protein in host adhesion. Whether the *sar5D*-encoded protein may be less efficient in host adhesion, due to the loss of two MucBP domains, remains

to be investigated. BPS was previously detected in a similar portion of invasive and colonizing isolates [37], suggesting the presence of the protein alone does not increase GBS' ability to invade/cause disease, for instance by enhancing host adhesion.

When BPS was first identified in 2002 by Erdogan *et al*, it was considered a new protein and different from R3 [15]. BPS was described in the reference strain NCTC 9828 (called Compton R by Erdogan *et al* [15]), which at that time was considered a Rib and R3 reference strain. However, later that same year, Kong *et al*. reported that the gene thought to encode Rib in strain NCTC 9828 (termed Prague 25/60 by Kong *et al*) had extensive similarities to the *rib* gene but also possessed stretches which differed from *rib* [9]. They named this new protein Alp4, which has been the designation used since then [16]. Later, it was reported that the C-terminal antigenic determinant of Alp4 and Rib cross-reacted immunologically, while the N-terminal antigenic determinants of Rib and Alp4 differed in immunological specificity [16]. The knowledge that NCTC 9828 carries *alp4* (and not *rib*) has consequences for the production of specific antisera targeting R3 and BPS in the study identifying BPS as a distinct protein from R3 [15]. Production of specific antibodies was performed by immunizing a rabbit with strain NCTC 9828. Harvested antiserum was adsorbed using a GBS strain expressing *rib*. This would remove antibodies targeting Rib, and is a common procedure for generating specific polyclonal antiserum. However, since NCTC 9828 does not express Rib, but the similar Alp4, antibodies targeting epitopes common to both Rib and Alp4 would be removed by the adsorption whereas antibodies specific only to Alp4 would remain in the antiserum. Immunoprecipitation-bands that were considered evidence of R3 by Erdogan *et al*. in 2002 [15], may therefore in fact have been bands representing Alp4. Data from both the present study with GBS strains from Norway as well as previous data from a collection of GBS strains from Zimbabwe [8] show a nearly perfect correlation between R3 protein expression and the presence of *sar5*. A study reporting 155 (of 4425 total) colonizing and invasive GBS strains expressing BPS found no overlap between R3- and BPS-expression [37]. The presumed R3-specific antibody used in that study was also prepared by adsorbing antisera made by immunizing a rabbit with GBS strain NCTC 9828. Regarding BPS and R3 as two distinct proteins would result in an R3-designated antibody without antibodies targeting the *sar5* gene product. Using NCTC 9828 as strain for R3 antibody production could therefore result in R3 antiserum that may actually detect Alp4. Because NCTC 9828 is the only GBS strain reported to have the *alp4* gene [16], generation of specific antibodies by adsorption to remove Alp4-targeting antibodies is not feasible using any known GBS strain. The BPS-specific antiserum is no longer available (Patricia Ferrieri, personal communication), and we were therefore not able to experimentally validate our speculations.

As a consequence of our findings, already reported data on BPS and R3 are equally relevant for the *sar5* gene product. Using BPS as the future designation of the *sar5* gene product makes the historical R3 protein nonexistent, and vice versa. The nomenclature of GBS surface proteins is already confusing (Table 1). BPS, R5, and now R3 are all names for the same protein. It is important that communication and reports use unambiguous terminology for genes and gene products. We therefore suggest using the designation R3/BPS for the *sar5*-encoded protein henceforth.

Existing data suggest that the prevalence of *sar5* in GBS strains differs between geographical regions. In Norway and the United States, the prevalence has been reported to lie in the range 2.3–8.1% in invasive GBS strains [20, 29, 37]. In a study from the United States, 3.6% of more than 4000 colonizing GBS strains carried R3/BPS [37], while in Zimbabwe it amounted to near 30% in healthy pregnant carriers [8]. Thus, as a strain variable marker, the R3/BPS protein has proved its potential in serotyping, as a serosubtype marker. Moreover, a recombinant version of this protein has been reported as immunogenic and, on immunization, induced formation

of antibodies protective in an animal model, suggesting potential for this protein as a vaccine component [15]. R3/BPS may thus be suitable as one of the constituents in a vaccine targeting GBS, particularly in vaccines aimed at populations in areas of Southern Africa where the presence of R3/BPS in GBS is high [8].

## Supporting information

**S1 Table. Overview of the R3 typing of the 140 GBS strains examined in this study, as determined by Kvam *et al*. [20], and the *sar5* typing performed in the current study (corresponding to the results of S1 Fig).**
(XLSX)

**S1 File. Primers used and illustration of the cloning strategies of pKT1-sar5_F and pKT1-sar5D_F.**
(PDF)

**S1 Fig. The *sar5* typing (PCR amplification) of the 140 GBS strains examined in this study. Negative (-) controls were strain 94–51 (a known R3 negative strain) or H$_2$O, positive (+) controls were the R3 reference strain CCUG 29784. A) amplification of the *sar5* gene from the 7 R3 positive GBS strains.** The black dashed line indicates that the image was edited at this location, removing one lane containing a sample that was excluded from the collection due to uncertainties around its identity. B-E) amplification of *sar5* from the 133 R3 negative strains, in pools of 3–5 strains. F) amplification of *sar5* from the strains within pool 7 and pool 9. Raw data images of electrophoresis gels are found in S1 Raw images.
(PDF)

**S2 Fig. Alignment of sequenced *sar5* PCR products of CCUG 29784, 93–33, and 94–3 to the *sar5* gene of the whole genome sequenced strain NCTC 9828.** Alignments were made using SnapGene software (from Insightful Science).
(PDF)

**S1 Raw images. Raw data images for Figs 1C, 4, and S1 Fig.**
(PDF)

## Acknowledgments

We thank the late Johan A. Mæland for the idea and encouragement for this project. Confocal microscopy was performed at the Cellular and Molecular Imaging Core Facility at NTNU.

## Author Contributions

**Conceptualization:** Christina Gabrielsen Aas, Jan Egil Afset, Marte Singsås Dragset.

**Data curation:** Adelle Basson, Camilla Olaisen, Linn-Karina Selvik, Randi Valsø Lyng, Hilde Lysvand, Alexandre Gidon, Christina Gabrielsen Aas, Jan Egil Afset, Marte Singsås Dragset.

**Formal analysis:** Adelle Basson, Linn-Karina Selvik, Randi Valsø Lyng, Hilde Lysvand, Christina Gabrielsen Aas, Jan Egil Afset, Marte Singsås Dragset.

**Funding acquisition:** Jan Egil Afset.

**Investigation:** Alexandre Gidon, Jan Egil Afset.

**Methodology:** Camilla Olaisen, Randi Valsø Lyng, Alexandre Gidon, Christina Gabrielsen Aas, Jan Egil Afset, Marte Singsås Dragset.

**Project administration:** Jan Egil Afset.

**Resources:** Jan Egil Afset.

**Software:** Alexandre Gidon, Christina Gabrielsen Aas, Jan Egil Afset.

**Supervision:** Christina Gabrielsen Aas, Jan Egil Afset, Marte Singsås Dragset.

**Validation:** Adelle Basson, Camilla Olaisen, Linn-Karina Selvik, Randi Valsø Lyng, Hilde Lysvand, Christina Gabrielsen Aas, Jan Egil Afset, Marte Singsås Dragset.

**Visualization:** Adelle Basson, Camilla Olaisen, Linn-Karina Selvik, Randi Valsø Lyng, Hilde Lysvand, Alexandre Gidon, Christina Gabrielsen Aas, Jan Egil Afset, Marte Singsås Dragset.

**Writing – original draft:** Adelle Basson, Camilla Olaisen, Alexandre Gidon, Christina Gabrielsen Aas, Jan Egil Afset, Marte Singsås Dragset.

**Writing – review & editing:** Adelle Basson, Camilla Olaisen, Hilde Lysvand, Alexandre Gidon, Christina Gabrielsen Aas, Jan Egil Afset, Marte Singsås Dragset.

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
