## [Decision Letter · Decision Letter 0]

16 Feb 2022

PONE-D-22-00654The Streptococcus agalactiae R3 surface protein is encoded by sar5PLOS ONE

Dear Dr. Dragset,

Thank you for submitting your manuscript to PLOS ONE. After careful consideration, we feel that it has merit but does not fully meet PLOS ONE’s publication criteria as it currently stands. Therefore, we invite you to submit a revised version of the manuscript that addresses the points raised during the review process.

We look forward to receiving your revised manuscript.

Kind regards,

Thomas Proft, Ph.D

Academic Editor

PLOS ONE

Journal Requirements:

"This work was partly supported by the Research Council of Norway through its Centres of Excellence scheme, project number 223255/F50."

Reviewers' comments:

Reviewer's Responses to Questions

**Comments to the Author**

1. Is the manuscript technically sound, and do the data support the conclusions?

Reviewer #1: Partly

Reviewer #2: Yes

2. Has the statistical analysis been performed appropriately and rigorously? 

Reviewer #1: N/A

Reviewer #2: N/A

3. Have the authors made all data underlying the findings in their manuscript fully available?

Reviewer #1: Yes

Reviewer #2: Yes

4. Is the manuscript presented in an intelligible fashion and written in standard English?

Reviewer #1: Yes

Reviewer #2: Yes

5. Review Comments to the Author

Reviewer #1: General comments:

The authors present an interesting study on the genetic background of the GBS surface protein R3. While for the vast majority of GBS surface proteins the genetic background has been elucidated, the gene encoding R3, which has been identified based on serologic detection and is frequently used for GBS subtyping, is unknown. Controversial reports and renaming of GBS surface proteins added to a situation that is not easy to unravel. In view of vaccine development and the potential use of these proteins in a future vaccine it is important to clarify the situation. In their study the authors could show that the sar5 gene encoding the BSP/R5 surface protein is also encoding the R3 protein that has mainly been defined through serologic studies. Based on the results of a previous investigation that showed a high correlation between the presence of the sar5 gene and the serologic detection of R3 the authors conducted a study on 140 Norwegian GBs strains. This study confirmed the correlation between sar5 and R3 detection. In addition the recombinant expression of sar5 in E. coli including truncated sar5 genes could be detected by R3 antibodies. One weekness of the manuscript is that among 140 strains only 7 were positive for the gene and reacted with the R3 antibody. 2 Strains carried truncated forms of sar5 and the authors conducted detailed investigation to show that these strains carried truncated forms of R3. However, the conclusion of the manuscript could be greatly substantiated through serologic studies showing that antiserum directed against BSP/R5 showed a crossreaction with the R3 positive strains, and with the recombinant proteins in E. coli. Their fear that existing antibodies due to preabsorption with not clearly defined GBS strains may recognize other surface proteins than intended can be ruled out through testing with the proteins in question. Another possibility would be to delete the sar5 gene in GBS and show that this leads to a loss of reactivity with R3 antibodies.

Specific comments:

1 Page 3 line 55

The authors state that 18% of women worldwide are colonized with GBS. Here a range instead of a fixed number for the colonization rate should be given, because the this rate varies quite a bit in different populations.

2 Page 5 line 113-115

The 140 Norwegian strains should also be tested with an antibody specific for BSP/R5 to substantiate the claim that these surface proteins are identical.

3 Page 6 line 128

Please specify if detection of the sar5 gene was confirmed through sequencing of the PCR product.

4 Page 14 line 315-327

The authors claim that older antibody preparations preabsorbed with strain NCTC 9828 may actually recognize Alp4 instead of the intended targets. This part is very speculative and should be tested experimentally.

Reviewer #2: I read with interest the paper by Dr. Marte Dragset et al. entitled "The Streptococcus agalactiae R3 surface protein is encoded by sar5" (Manuscript Number PONE-D-22-00654). The paper convincingly shows that the R3 protein is encoded by sar5 and the authors propose the use of the R3/BPS designation for unambiguously indicating the sar5 gene product. The finding has useful practical implications for the serotyping of an important pathogen, such as Streptococcus agalactiae.

Major comments

1. The abstract should be rewritten to more clearly and thoroughly convey the results of the study. The introductory part in the abstract consists of 7 of the 8 total lines. I suggest that the authors reduce the introductory part to one or two sentences and take advantage of the full allowed length of the abstract to illustrate the results and the conclusions.

2. The paper would benefit from provision of a simple, schematic representation of the protein. The authors should try to identify putative domains by bioinformatic analysis using programs such as Pfam. For example the presence/absence or a signal peptide should be identified because it is relevant to the findings.

3. The figure provided (Fig. 2), concerning gene alignment, should be improved because it is difficult to read and out of focus

4. Although only limited information is available in the literature concerning the biological function of the R3/BPS protein, the authors should discuss this point. This is crucial in order to properly discuss results, particularly those dealing with the observation that partial gene deletion results in lack of surface expression. Is it likely that the portion of the protein lacking in the truncated form is important for secretion?

5. Ideally, the authors should try to express the truncated and non-truncated forms of the gene in a GBS strain lacking sar5 and verify experimentally whether the gene product is present on the bacterial surface. This would make the paper more appealing from a biological perspective. However, this is not absolutely necessary to support the conclusions of the present study, which is focused on serotyping.

6. There are few typing mistakes:

lines 28 and 53, not "a group", but "group"

line 80: do you mean "Ca-like proteins 1-4 (Alps 1-4)"?

line 144 not "has been deletion ", but "has been deleted"

line185 not "shown to possessed", but "shown to possess"

6. PLOS authors have the option to publish the peer review history of their article (what does this mean?). If published, this will include your full peer review and any attached files.

Reviewer #1: No

Reviewer #2: **Yes: **Concetta Beninati

---

## [Author Response · Author response to Decision Letter 0]

4 May 2022

To the Reviewers of PLOS ONE submission PONE-D-22-00654

We would like to sincerely thank you for carefully reading and commenting on our manuscript, PONE-D-22-00654 “The Streptococcus agalactiae R3 surface protein is encoded by sar5”. Below is a point-by-point response to your constructive critiques regarding our results and conclusions. Reviewer’s comments are shown in italics. We have addressed the criticism by clarifying the writing and including new figures from experiments and in silico analysis. The changes have been incorporated in our revised manuscript, as detailed below and in the “Revised Manuscript with Track Changes”: 

Reviewer #1: General comments:

The authors present an interesting study on the genetic background of the GBS surface protein R3. While for the vast majority of GBS surface proteins the genetic background has been elucidated, the gene encoding R3, which has been identified based on serologic detection and is frequently used for GBS subtyping, is unknown. Controversial reports and renaming of GBS surface proteins added to a situation that is not easy to unravel. In view of vaccine development and the potential use of these proteins in a future vaccine it is important to clarify the situation. In their study the authors could show that the sar5 gene encoding the BSP/R5 surface protein is also encoding the R3 protein that has mainly been defined through serologic studies. Based on the results of a previous investigation that showed a high correlation between the presence of the sar5 gene and the serologic detection of R3 the authors conducted a study on 140 Norwegian GBs strains. This study confirmed the correlation between sar5 and R3 detection. In addition the recombinant expression of sar5 in E. coli including truncated sar5 genes could be detected by R3 antibodies. One weekness of the manuscript is that among 140 strains only 7 were positive for the gene and reacted with the R3 antibody. 2 Strains carried truncated forms of sar5 and the authors conducted detailed investigation to show that these strains carried truncated forms of R3. However, the conclusion of the manuscript could be greatly substantiated through serologic studies showing that antiserum directed against BSP/R5 showed a crossreaction with the R3 positive strains, and with the recombinant proteins in E. coli. Their fear that existing antibodies due to preabsorption with not clearly defined GBS strains may recognize other surface proteins than intended can be ruled out through testing with the proteins in question. Another possibility would be to delete the sar5 gene in GBS and show that this leads to a loss of reactivity with R3 antibodies.

We agree that the low number (7) of R3 positive strains in our collection might represent a weakness, however, this is merely a consequence of the intrinsic low prevalence of R3 in strains collected in Norway compared to those collected from for instance Zimbabwe. The Norwegian strain collection was the only one available to us for this study. Even so, with relatively few R3 positive strains at hand, we were able to pinpoint a correlation between R3 and the sar5 gene. We answer suggestions regarding testing the BPS antiserum on our strain collection and our speculations around the BPS/R5 antiserum’s specificity further below in the response letter. 

Specific comments:

1 Page 3 line 55

The authors state that 18% of women worldwide are colonized with GBS. Here a range instead of a fixed number for the colonization rate should be given, because the this rate varies quite a bit in different populations.

We agree. We used a 2016 meta-analysis published in Lancet Infectious Diseases including data from over 70,000 women in 37 countries (doi: 10.1016/S1473-3099(16)30055-X) as our reference for global GBS prevalence. However, we failed to communicate clearly that the 18% was based on a world estimate/mean. We have clarified this by re-writing this paragraph and included the range of highest and lowest prevalence found in this study within the different populations (line 55 page 2): “Globally, an estimated prevalence of maternal rectovaginal GBS colonization is 17.9 %, with the highest and lowest mean prevalence found in Africa (22.4%) and Southeast Asia (11.1%), respectively ».

2 Page 5 line 113-115

The 140 Norwegian strains should also be tested with an antibody specific for BSP/R5 to substantiate the claim that these surface proteins are identical.

We agree. Showing that the BPS/R5 antiserum cross-reacts with R3 positive strains only would be the ultimate way to prove that BPS/R5 and R3 are indeed one and same protein. Therefore, we contacted authors from the BPS/R5 publications (Erdogan et al, Infect Immun 2002;70:80, Flores et al Curr Microbiol 2014;69:894) already back in June 2021 and received the answer that this antiserum is no longer available. For this, we are unfortunately not able to perform the suggested experiment. We have attached this email correspondence at the end of this resubmission letter (Appendix I). 

3 Page 6 line 128

Please specify if detection of the sar5 gene was confirmed through sequencing of the PCR product.

We had not sequenced the sar5 PCR product and understand the reviewer’s concern that we may indeed have amplified something unspecific and unrelated to sar5. Therefore, we sequenced the PCR products arising from amplification with primers specific to the 5’ end of the sar5 gene (the same primers that we used in the manuscript, from Mavenyengwa et al (DOI: 10.4103/0255-0857.71819)) for three representative strains; 1) the positive control strain (CCUG 29784), 2) one of the R3-negative sar5-positive strains (93-33) and one R3-postive sar5-positive strains (93-50). All sequences aligned to the sar5 gene of the genome sequenced strain NCTC 9828. 

To update the manuscript accordingly, we added the text “For three representative strains (the R3 reference strain CCUG 29784, the R3-negative sar5-positive strain 93-33, and the R3-positive sar5-positive strain 93-50) we confirmed by Sanger sequencing that the PCR products indeed represented amplification of the sar5 gene (S2 Figure)” at line 122 page 6 in the results section, and “; sar5 forward and sar5 reverse (8). For three representative strains (CCUG 29784 as the positive control strain, 93-33 as an R3-negative sar5-positive strain, and 93-50 as an R3-positive sar5-positive strain) the PCR products were sequenced using the sar5 forward primer and the Eurofins Genomics sequencing services (LightRun Tube format).” at line 266 page 12 in materials and methods. We added the Supplementary figure S2 Figure and updated the figure legend accordingly. 

4 Page 14 line 315-327

The authors claim that older antibody preparations preabsorbed with strain NCTC 9828 may actually recognize Alp4 instead of the intended targets. This part is very speculative and should be tested experimentally.

We agree. While BPS was previously identified as the protein encoded by sar5 and different from R3, we demonstrate that R3 is also encoded by sar5. It is therefore crucial to our presented work that we properly discuss why we think BPS and R3 were first identified as different proteins. Our speculations around this should ideally be tested experimentally, however, because the BPS-specific antiserum is no longer available we are unable to do so. As we experimentally and unambiguously show that sar5 encodes R3, we find it out of the scope of this work to re-create the BPS antiserum. To clarify this to the reader, we added the following sentence to the discussion (line 414 page 18): “The BPS-specific antiserum is no longer available (Patricia Ferrieri, personal communication), and we were therefore not able to experimentally validate our speculations.”. We also added references and text to this paragraph to make it clearer to the reader, starting at line 402 page 17.

Reviewer #2: I read with interest the paper by Dr. Marte Dragset et al. entitled "The Streptococcus agalactiae R3 surface protein is encoded by sar5" (Manuscript Number PONE-D-22-00654). The paper convincingly shows that the R3 protein is encoded by sar5 and the authors propose the use of the R3/BPS designation for unambiguously indicating the sar5 gene product. The finding has useful practical implications for the serotyping of an important pathogen, such as Streptococcus agalactiae.

Major comments

1. The abstract should be rewritten to more clearly and thoroughly convey the results of the study. The introductory part in the abstract consists of 7 of the 8 total lines. I suggest that the authors reduce the introductory part to one or two sentences and take advantage of the full allowed length of the abstract to illustrate the results and the conclusions.

We agree and have updated the abstract accordingly (line 37 page 2): “, like the R3 surface protein, are still unknown. Here, by examining a Norwegian GBS collection consisting of 140 strains, we find that R3 protein expression correlates with the presence of the gene sar5. By inducible expression of sar5 in an R3-negative bacterial strain we show that the sar5 gene product is specifically recognized by an R3 monoclonal antibody. With this we identify sar5 as the gene encoding the R3 surface protein, a serosubtype marker of hitherto unknown genetic origin.”

2. The paper would benefit from provision of a simple, schematic representation of the protein. The authors should try to identify putative domains by bioinformatic analysis using programs such as Pfam. For example the presence/absence or a signal peptide should be identified because it is relevant to the findings.

This is an interesting and important point raised by the reviewer that we had not considered. When we ran the amino acid sequences of the sar5 gene products from NCTC 9828 and strains 93-33 and 94-3 through the Pfam database, we found that the deleted region corresponds to two mucin-binding protein (MucBP) domains but that both the signal peptide and cell wall anchor motif remain intact. MucBPs are a group of surface proteins that are thought to facilitate adhesion of bacteria to the host via interactions with the mucins of the mucus secreted by epithelial cells (DOI: 10.1016/j.jsb.2010.10.016, DOI:10.1111/1462-2920.12377). Our new findings may point in the direction of R3 being involved in GBS adhesion to the host. We created a new figure representing these findings (new Figure 2B), updating the figure legend accordingly. We also added this text to the results section (line 203 page 7): “To investigate if the deletion was likely to interfere with surface expression, we subjected the sar5-encoded protein sequences of strains NCTC 9828 and 93-33/94-3 to the Pfam database (https://pfam.xfam.org/) for domains/motifs prediction. We found that the deleted region corresponds to two of five mucin-binding protein domains (MucBP; PF06458) found in the full-length protein (Figure 2B). Moreover, a signal peptide (YSIRK_signal; PF04650) and a cell wall anchor motif (LPXTG gram_pos_anchor; PF00746) were predicted at the N- and C-terminus of the sar5 gene products, respectively, for both NCTC 9828 and 93-33/94-3 (Figure 2B). These findings suggest that the deletion does not directly interfere with surface expression”. We updated the legend for Figure 2B accordingly. 

3. The figure provided (Fig. 2), concerning gene alignment, should be improved because it is difficult to read and out of focus.

We agree and have improved the quality of Figure 2 on the gene alignment, including increased font sizes. We also removed old Figure 2B and replaced it with the Pfam predictions, as we saw it was not adding substantially to the research story. 

4. Although only limited information is available in the literature concerning the biological function of the R3/BPS protein, the authors should discuss this point. This is crucial in order to properly discuss results, particularly those dealing with the observation that partial gene deletion results in lack of surface expression. Is it likely that the portion of the protein lacking in the truncated form is important for secretion?

We agree. However, we need to underline that we never directly observed that the gene deletion resulted in lack of surface expression, we merely speculated around this. We address these speculations in more detail in the comment below. Even so, the observations we made during the bioinformatical analysis of our proteins (comment 2) shed some light on the possible biological function of R3/BPS, while at the same time suggesting that the deletion does not interfere with secretion nor cell wall anchoring. From this we included the following paragraph in the discussion (line 371 page 16): “Bioinformatic prediction of the sar5D gene product revealed that its signal peptide and cell wall anchoring motif were not affected by the deletion (Figure 2B), providing no direct explanation of the potential lack of surface expression. Predictions also revealed that the sar5D deleted region corresponds to two of five MucBP domains. MucBP domains are thought to facilitate adhesion of bacteria to the host via direct interactions with mucins within epithelial cell-secreted mucus (36, 37), shedding light on a possible function of the sar5-encoded protein in host adhesion. Whether the sar5D-encoded protein may be less efficient in host adhesion, due to the loss of two MucBP domains, remains to be investigated. BPS was previously detected in a similar portion of invasive and colonizing isolates (38), suggesting the presence of the protein alone does not increase GBS’ ability to invade/cause disease, for instance by enhancing host adhesion.”

5. Ideally, the authors should try to express the truncated and non-truncated forms of the gene in a GBS strain lacking sar5 and verify experimentally whether the gene product is present on the bacterial surface. This would make the paper more appealing from a biological perspective. However, this is not absolutely necessary to support the conclusions of the present study, which is focused on serotyping.

We agree. The localization of the sar5 deletion mutant is interesting from a biological perspective, while not directly relevant for the presented research. Even so, we subjected GBS strains possessing either the sar5 or the sar5D genes to confocal microscopy to investigate the R3/BPS localization. By using the clinical strains already available we avoided potential problems that might have arisen from vector-based expression. While the cells were too small to specifically localize R3 on the surface of the cells we were able to quantify R3 antibody-mediated signal from stacks of images and normalize these to the signal from a dye binding proteins in general (CPD eFluor 450). From this, we clearly saw that the R3 signal from the two sar5D strains were at the same level as the negative control. This may suggest that the truncated R3 encoded by sar5D is not expressed on the cell surface, however, bioinformatical analysis showed that its signal peptide is intact. It therefore inconclusive whether this protein is expressed on the surface or not, and we consider it out of the scope to further pursue the biological function of the sar5/sar5D gene product for this study. Our new confocal analysis adds value to our work though, by validating that the truncated R3 does not bind the R3 antibody in a whole cell state, confirming previous data on the same strains from reference (18). For this, we added the following paragraph to the results section (line 136 page 6): “To re-ensure that these two strains are indeed not recognized by the R3 antibody in a whole cell state, we R3-stained and subjected them to confocal microscopy together with the R3/sar5 negative (CCUG 29779) or positive (CCUG 29784) control strains (Figure 1A). By quantifying their R3 antibody signals, the 93-33 and 94-3 strains showed the same signal level as the negative control (Figure 1B)”, and updated the figure legend accordingly. We also added two new paragraph in the materials and methods describing the sample preparations and confocal microscopy methodology and included co-author Alexandre Gidon for conducting the confocal experiment. Finally, to make the second paragraph in the results section flow with the new results introduced, changes were made throughout the paragraph, starting at line 143 page 7.

6. There are few typing mistakes:

lines 28 and 53, not "a group", but "group"

 – corrected

line 80: do you mean "Ca-like proteins 1-4 (Alps 1-4)"?

 – We believe the original sentence is correct, as Ca is one of the Ca-like proteins (Alps) (see Table 1). 

line 144 not "has been deletion ", but "has been deleted" 

– corrected

line185 not "shown to possessed", but "shown to possess" 

– corrected

APPENDIX I

Mail correspondence between Prof. Jan Egil Afset and Prof. Patricia Ferrieri regarding the BPS/R5 antiserum. 

Request: 

From: Jan Egil Afset 

Sent: Tuesday, June 16, 2020 12:09 PM

To: 'ferri002@umn.edu' <ferri002@umn.edu>

Subject: BPS rabbit antiserum

Dear prof. Ferrieri,

The undersigned is a professor in microbiology at the Norwegian University of Science and Technology and St. Olavs hospital, which harbors the national reference laboratory for GBS in Norway. In relation to the reference laboratory we do typing of invasive GBS strains both with respect to CPS- and surface protein type. We have antisera for most CPS- and surface proteins, and have had some focus on R3 through the years. Would it be possible to get some of the rabbit BPS antiserum that you have developed (Erdogan et al, Infect Immun 2002;70:80, Flores et al Curr Microbiol 2014;69:894) for comparison between our R3-antiserum?

With best regards,

 Jan Egil Afset

Jan Egil Afset, MD, PhD

Professor, Department of Clinical and Molecular Medicine, NTNU

Consultant, Department of Medical Microbiology, St. Olavs Hospital

Trondheim, Norway

https://www.ntnu.no/ansatte/jan.afset

Answer: 

From: Patricia Ferrieri <ferri002@umn.edu> 

Sent: Tuesday, June 30, 2020 5:26 PM

To: Jan Egil Afset <jan.afset@ntnu.no>

Subject: Re: BPS rabbit antiserum

Dear Professor Afset,

I regret that I do not have any of this antiserum left to help you.

Best regards,

Patricia Ferrieri

Patricia Ferrieri, M.D.

Professor

Chairman's Fund Endowed Chair in Laboratory Medicine and Pathology

Professor, Dept of Pediatrics, Division of Infectious Diseases

University of Minnesota Medical School

Director, Infectious Diseases Diagnostic Laboratory

University of Minnesota Medical Center

Mayo Mail Code 134

420 Delaware St. SE Minneapolis, MN 55455

612-626-6645 (FAX)

---

## [Decision Letter · Decision Letter 1]

1 Jun 2022

The Streptococcus agalactiae R3 surface protein is encoded by sar5

PONE-D-22-00654R1

Dear Dr.  Singsås Dragset,

We’re pleased to inform you that your manuscript has been judged scientifically suitable for publication and will be formally accepted for publication once it meets all outstanding technical requirements.

Kind regards,

Thomas Proft, Ph.D

Academic Editor

PLOS ONE

**Comments to the Author**

1. If the authors have adequately addressed your comments raised in a previous round of review and you feel that this manuscript is now acceptable for publication, you may indicate that here to bypass the “Comments to the Author” section, enter your conflict of interest statement in the “Confidential to Editor” section, and submit your "Accept" recommendation.

Reviewer #1: All comments have been addressed

Reviewer #2: All comments have been addressed

2. Is the manuscript technically sound, and do the data support the conclusions?

Reviewer #1: (No Response)

Reviewer #2: Yes

3. Has the statistical analysis been performed appropriately and rigorously? 

Reviewer #1: (No Response)

Reviewer #2: Yes

4. Have the authors made all data underlying the findings in their manuscript fully available?

Reviewer #1: (No Response)

Reviewer #2: Yes

5. Is the manuscript presented in an intelligible fashion and written in standard English?

Reviewer #1: (No Response)

Reviewer #2: Yes

6. Review Comments to the Author

Reviewer #1: (No Response)

Reviewer #2: (No Response)

7. PLOS authors have the option to publish the peer review history of their article (what does this mean?). If published, this will include your full peer review and any attached files.

Reviewer #1: No

Reviewer #2: No

---

## [Editor Report · Acceptance letter]

8 Jul 2022

PONE-D-22-00654R1 

The *Streptococcus agalactiae* R3 surface protein is encoded by *sar5*

Dear Dr. Dragset:

I'm pleased to inform you that your manuscript has been deemed suitable for publication in PLOS ONE. Congratulations! Your manuscript is now with our production department. 

Kind regards, 

on behalf of

Dr. Thomas Proft 

Academic Editor

PLOS ONE